# CBL-Interacting Protein Kinase 2 Improves Salt Tolerance in Soybean (*Glycine max* L.)

Hui Li [1,2], Zhen-Ning Liu [1], Qiang Li [1], Wen-Li Zhu [3], Xiao-Hua Wang [1], Ping Xu [1], Xue Cao [1] and Xiao-Yu Cui [1,*]

[1]   College of Agriculture and Forestry Sciences, Linyi University, Linyi 276000, China; lihuiqau@163.com (H.L.); liuzhenning@lyu.edu.cn (Z.-N.L.); liqiang@lyu.edu.cn (Q.L.); wangxiaohua@lyu.edu.cn (X.-H.W.); xuping@lyu.edu.cn (P.X.); caoxue@lyu.edu.cn (X.C.)
[2]   Center for International Education, Philippine Christian University, Metro Manila 1004, Philippines
[3]   College of Modern Agriculture, Linyi Vocational University of Science and Technology, Linyi 276000, China; wlzhu22@126.com
*   Correspondence: cuixiaoyu@lyu.edu.cn

**Abstract:** Salt stress severely limits soybean production worldwide. Calcineurin B-like protein-interacting protein kinases (CIPKs) play a pivotal role in a plant's adaption to salt stress. However, their biological roles in soybean adaption to salt stress remain poorly understood. Here, the *Gm-CIPK2* expression was increased by NaCl and hydrogen peroxide ($H_2O_2$). *GmCIPK2*-overexpression *Arabidopsis* and soybean hairy roots displayed improved salt tolerance, whereas the RNA interference of hairy roots exhibited enhanced salt sensitivity. Further analyses demonstrated that, upon salt stress, *GmCIPK2* enhanced the proline content and antioxidant enzyme activity and decreased the $H_2O_2$ content, malondialdehyde (MDA) content, and $Na^+/K^+$ ratios in soybean. Moreover, *GmCIPK2* promoted the expression of salt- and antioxidant-related genes in response to salt stress. Moreover, the GmCIPK2-interacting sensor, GmCBL4, increased the salt tolerance of soybean hairy roots. Overall, these results suggest that *GmCIPK2* functions positively in soybean adaption to salt stress.

**Keywords:** soybean; CIPK; salt tolerance





## 1. Introduction

Plants are sessile organisms that often encounter various environmental changes, including salt, drought, and extreme temperatures [1,2]. During evolution, plants have evolved complex strategies to adapt to unfavorable conditions [2,3]. Calcium ($Ca^{2+}$) is a universal secondary messenger that regulates plant growth, development, and stress responses [4,5]. Environmental stimuli can trigger spatio-temporal changes in cytoplasmic $Ca^{2+}$ concentrations [4,6]. The changes are then detected by $Ca^{2+}$ sensors, such as $Ca^{2+}$-dependent protein kinases, calmodulins, and calcineurin B-like proteins (CBLs) [6,7]. Subsequently, the $Ca^{2+}$ sensors interact with their downstream targets, causing a series of physiological and metabolic alterations in plants [7,8].

CBLs can specifically bind to CBL-interacting protein kinases (CIPKs) to form plant-specific $Ca^{2+}$ signal decoding systems [4,7,9]. CIPKs have been identified as a class of serine/threonine (Ser/Thr) kinases that share a close evolutionary relationship with SNF1 (sucrose non-fermenting-1)-related kinases 3 [7,8]. The catalytic domain of CIPKs is located at the N-terminus and contains an activation loop and an ATP binding site [7,9]. A NAF/FISL motif is found in the C-terminus of CIPKs, adjacent to the junction domain [10]. The NAF/FISL motif is critical for the interaction with CBLs [8,9]. Since the CBL-CIPK pathway was discovered in *Arabidopsis* (*Arabidopsis thaliana* L.) [11], CIPK homologs have been found in rice (*Oryza sativa* L.) [12], pepper (*Capsicum annuum* L.) [13], maize (*Zea mays* L.) [14], tomato (*Solanum lycopersicum* L.) [15], soybean (*Glycine max* L.) [16], wheat (*Triticum aestivum* L.) [17], apple (*Malus domestica* Borkh.) [18], and cotton (*Gossypium hirsutum* L.) [19].

CIPKs affect ion homeostasis, hormonal signaling, and tolerance to abiotic stresses [4,9]. For example, *AtCIPK8* has been demonstrated to positively regulate *Arabidopsis'* salt tolerance [20]. *AtCIPK24* is a key component of the salt overly sensitive pathway that regulates the salt tolerance of *Arabidopsis* [21]. Additionally, it has been demonstrated that *AtCIPK14* participates in *Arabidopsis'* glucose response [22]. Studies have explored the functions of CIPKs in other plant species. For instance, cold stress leads to the increased expression of *OsCIPK3* and *OsCIPK7*, and transgenic plants overexpressing (OE) *OsCIPK3* and *OsCIPK7* display cold-tolerant phenotypes [23,24]. Furthermore, *ZmCIPK16* regulates stress-responsive gene expression to mediate maize adaption to salt stress [25]. *TaCIPK27* and *TaCIPK23* respond to drought stress by regulating the stomatal movement [26,27]. In addition, *CaCIPK13* expression is induced by cold stress, and *CaCIPK13*-OE tomato plants exhibit cold-resistant phenotypes [28]. *SlCIPK24* is found to modulate $Na^+/K^+$ homeostasis in tomato salt responses [15].

Soybean is an important economic crop and a crucial source of edible oil, high-quality protein, and industrial products [29,30]. Salt stress is a major environmental challenge that severely restricts crop quality and yields worldwide [1,31]. It has been established that CIPKs play a crucial role in a plant's adaption to adverse conditions [4,5]. Nevertheless, whether CIPKs participate in alleviating salt stress in soybean remains largely unknown. Our previous study demonstrated that *GmCIPK2* serves as a positive regulator of drought tolerance for soybean [32]. In the present study, the biological functions of *GmCIPK2* in the salt response are characterized. Salt stress increases the transcript level of *GmCIPK2*. Further physiological and molecular assays demonstrate that *GmCIPK2* contributes to the salt tolerance of soybean.

## 2. Materials and Methods

### 2.1. Plant Materials and Growth Conditions

Soybean seedlings (Williams 82) were cultured in a growth room under a 16 h light/8 h dark photoperiod, in 70% relative humidity, at a temperature of 25 °C. Fourteen-day-old soybean plants of the same size were transferred to the 1/2 Hoagland's solution containing 200 mM NaCl and 10 mM $H_2O_2$ for the expression profile analysis. The soybean leaves that were used for the RNA extraction were harvested at 0, 1, 3, 7, 12, and 24 h. Seeds of *Arabidopsis* ecotype Columbia-0 were germinated in the 1/2 Murashige and Skoog (MS) medium in an illumination incubator with a photoperiod of 16 h light/8 h dark, in a relative humidity of 70%, at 23 °C. Twenty seeds were sown on each plate and vernalized for three days at 4 °C.

### 2.2. Transgenic Arabidopsis and Soybean Plant Construction

*Arabidopsis* with *GmCIPK2* overexpression was constructed by a previously described floral dip method [31,32]. Transgenic soybean hFairy roots were constructed by an *Agrobacterium rhizogenes*-mediated transformation, as described previously [1,30,33]. To obtain the OE transformation vectors, the full-length open reading frames of *GmCIPK2* and *GmCBL4* were inserted into pCAMBIA3301, driven by *cauliflower mosaic virus* 35S promoter, respectively. The sense and antisense fragments of *GmCIPK2* (28 bp–178 bp) were connected by the intron 6 of the rice zinc finger gene to constitute the specific RNA interference (RNAi) fragment [31,32]. The specific RNAi fragments were ligated into pCAMBIA3301 to generate the pCAMBIA3301-*GmCIPK2*-RNAi constructs. Subsequently, these vectors were transformed into *A. rhizogenes* strain K599. The 5-day-old soybean seedling was infected with *A. rhizogenes* strain K599, harboring the transformation vectors (RNAi, VC, and OE) around the cotyledonary node area with a syringe needle. The infected plants were then covered with plastic cups and kept in the dark at 28 °C. After 24 h, the plastic cups were removed. Meanwhile, the infection sites were covered with wet vermiculite until the hairy roots were generated. Two weeks later, the original roots of the soybean were removed, and then soybean plants were grown with hairy roots forming transgenic soybean hairy root composite plants. The transcript levels of functional genes were detected by qRT-PCR

assays. The composite plants were planted in flowerpots (12 cm × 14 cm) containing nutrient soil and vermiculite (1:1). Each pot contained 10 independent composite plants as a sample.

### 2.3. Quantitative Real-Time-PCR Assay

Total RNA was isolated and extracted using the RNA extraction kit (ZP401, Zomanbio, Beijing, China). The quantitative real-time (qRT) PCR analyses were conducted using an Applied Biosystems real-time PCR system and a TransStart Top Green qPCR SuperMix kit (AQ131, TransGen, Beijing, China) [31]. The relative expression levels of these selected genes were calculated with the $2^{-\Delta\Delta CT}$ method, and the *Gmtubulin* expression was used as an internal reference. The primers used for the qRT-PCR assay are displayed in Table S1.

### 2.4. Salt Tolerance Assay

The *Arabidopsis* seedlings were grown in the 1/2 MS medium for seven days. Then, these plants underwent salt treatment (75 mM NaCl). After 10 days, the total root length, proline content, malondialdehyde (MDA) content, and $H_2O_2$ content were measured. To analyze the soybean salt tolerance, the composite soybean plants were cultured in flowerpots (12 cm × 14 cm) containing nutrient soil and vermiculite (1:1) for one week. For the salt stress treatment, 0.4 L of NaCl solution (200 mM) was added to the bottom tray of each flowerpot once every 3 days. After treatment for 10 days, clearly wilting differences were distinguished between the transgenic (RNAi and OE) and control plants. Each sample contained 10 independent seedlings, and the experiments were repeated three times. For the analysis of the physiological parameters, soybean seedlings underwent the salt treatment for seven days. The transgenic soybean hairy roots were then collected to assay the proline content, MDA content, $H_2O_2$ content, peroxidase (POD), and glutathione S-transferase (GST) activity using the corresponding detection kit (BC3595, BC0025, BC0095, BC0355, Solarbio, Beijing, China). The contents of $Na^+$ and $K^+$ were analyzed by an inductively coupled plasma-optical emission spectrometer (ICP-OES, United States), as described previously [1].

### 2.5. Yeast Two-Hybrid Assay

Using the Matchmaker™ Two-Hybrid System, empty pGADT7 (AD), empty pGBKT7 (BD), *GmCBL4*-AD, and *GmCIPK2*-BD plasmids were transfected into yeast cells (*AH109*) according to the manufacturer's protocol. Subsequently, these transformants were plated on the SD/-Ade/-Leu/-Trp/-His medium containing X-α-gal [26,32].

### 2.6. Pull-Down Assay

To obtain the GmCIPK2-His and GmCBL4-GST recombinant proteins, pCold-*GmCIPK2* and pGEX-4T-1-*GmCBL4* were constructed and transfected into *Escherichia coli* (*BL21*), respectively. The glutathione agarose beads bounded with GmCBL4-GST proteins were then used to combine with the soluble GmCIPK2-His protein. The products were then washed and used for the Western blotting assay, as described previously [2,26].

### 2.7. Subcellular Localization Assay

The *GmCIPK2*-GFP and *GmCBL4*-mCherry plasmids were transformed into *Arabidopsis* protoplasts using a PEG-mediated transformation system. After incubating in the dark for 12 h, the fluorescence signal in the transfected protoplasts was analyzed by a confocal laser-scanning microscope [3,32].

### 2.8. Statistical Analysis

All experiments were repeated three times independently. The values are displayed as the mean ± SE of three biological replicates. The differences between the various treatments were analyzed using the one-way analysis of variance (ANOVA) using the SPSS software

(SPSS, statistics). Significant differences were determined by a Student's *t*-test and labeled as * $p < 0.05$.

## 3. Results

### 3.1. Isolation of Salt Stress-Responsive Gene GmCIPK2

CIPKs function essentially in plant tolerance to environmental stresses, while the functions of soybean CIPKs' tolerance to salt stress remain largely unknown. The BlASTP and multiple sequence alignment analyses demonstrated that GmCIPK2 has a high sequence identity with OsCIPK2 and AtCIPK2 (https://blast.ncbi.nlm.nih.gov/Blast.cgi, accessed on 25 April 2022). The structure analysis result revealed that GmCIPK2 contained typical CIPK domains: the N-terminal Ser/Thr kinase domain and the NAF/FISL domain (Figure S1). Further qRT-PCR assays demonstrated that NaCl-mediated salt stress increased the transcript levels of *GmCIPK2*, peaking at 3 h (Figure 1A), implying a potential role of *GmCIPK2* in salt response. Notably, $H_2O_2$-mediated oxidative stress led to an enhanced *GmCIPK2* expression with a similar expression pattern in response to salt stress (Figure 1B).

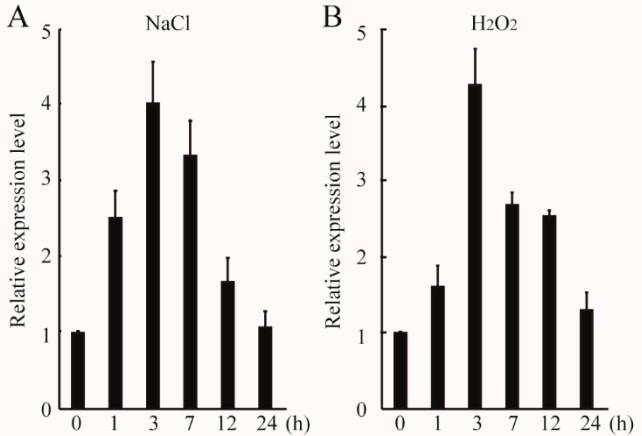

**Figure 1.** Transcript levels of *GmCIPK2* under salt and oxidative stresses. The expression levels of *GmCIPK2* under (**A**) NaCl treatment and (**B**) $H_2O_2$ treatment were measured in qRT-PCR assays. *Gmtubulin* was used as an internal reference. Each data point represents the mean ($\pm$SE) of three independent biological replicates.

### 3.2. GmCIPK2 Overexpression Confers Transgenic Arabidopsis Tolerance to Salt Stress

To explore the salt resistance associated with *GmCIPK2*, transgenic *Arabidopsis* plants with *GmCIPK2* overexpression (*GmCIPK2*-#3, *GmCIPK2*-#7, and *GmCIPK2*-#11) were generated (Figure 2B). Under favorable conditions, the OE *Arabidopsis* plants exhibited a similar phenotype to the wild-type (WT) plants. Nevertheless, the salt treatment resulted in significant differences in physiological traits among the different genotypes. The OE *Arabidopsis* plants showed salt-tolerant phenotypes with larger biomass accumulation and longer root lengths (Figure 2A,C,D). Proline is a well-characterized osmolyte that enhances plant tolerance to salt stress [3,26]. The salt-treated OE *Arabidopsis* plants accumulated a significantly larger proline content than the control plants (Figure 2E). Additionally, salt stress promoted malondialdehyde (MDA) synthesis, which was negatively correlated with salt tolerance [3]. The MDA levels of salt-treated OE *Arabidopsis* plants were lower than that of the control plants (Figure 2F).

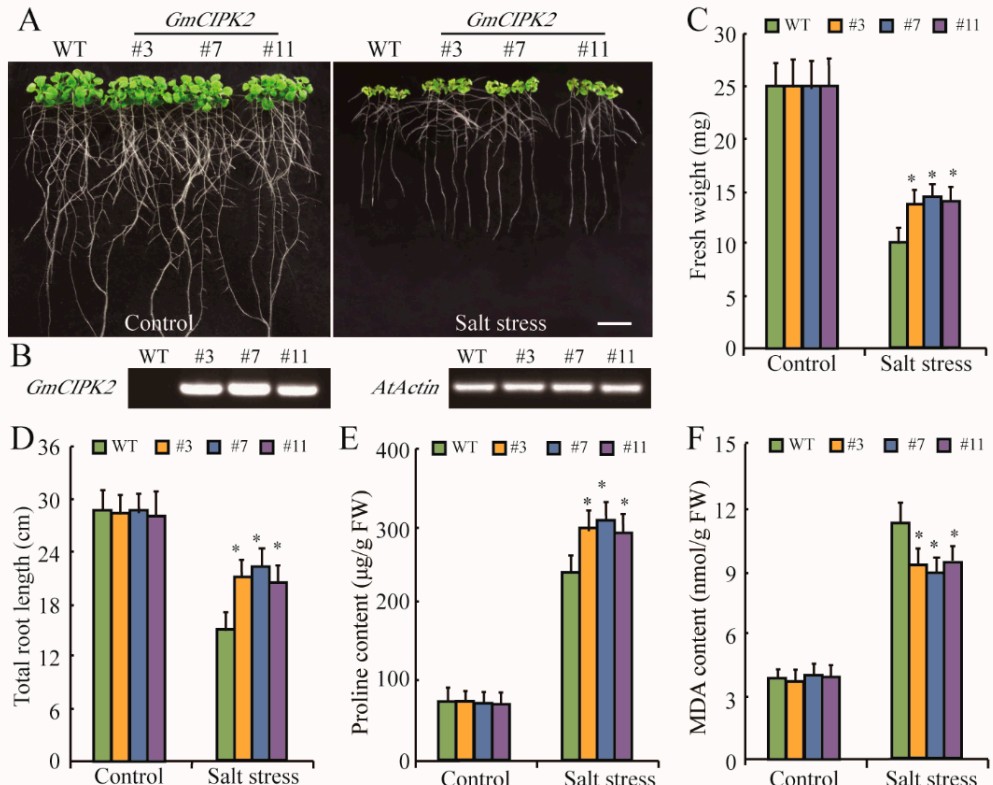

**Figure 2.** *GmCIPK2* overexpression results in transgenic-*Arabidopsis*-enhanced salt tolerance. (**A**) Analyses of the salt tolerance in OE and WT *Arabidopsis* plants. Bar = 1 cm. (**B**) The transcript of *GmCIPK2* was determined using semi RT-PCR assays. (**C**) Fresh weight, (**D**) total root length, (**E**) proline contents, and (**F**) MDA contents in OE and WT *Arabidopsis* plants under salt conditions. Each data point represents the mean (±SE) of three independent biological replicates. The * represents significant differences with the corresponding controls (* $p < 0.05$).

### 3.3. GmCIPK2 Promotes the Salt Tolerance of Soybean Hairy Roots

We generated transgenic hairy roots through the RNAi and OE technologies to verify the salt resistance associated with *GmCIPK2* in soybean. *GmCIPK2* transcript levels in transgenic hairy roots were examined using qRT-PCR assays (Figure 3B). Before salt stress, the different genotypes have no significant difference (Figure 3A,C). Subsequent salt stress triggers the phenotypic and physiological changes among the different genotypes. Under salt stress, the OE soybean plants exhibited better salt-tolerant phenotypes and higher survival rates than the vector control (VC) plants. In contrast, the RNAi soybean plants had salt-sensitive phenotypes with lower survival rates (Figure 3A,C). Furthermore, upon salt stress, the proline content was higher in the OE hairy roots than in the VC hairy roots. Conversely, the salt-treated RNAi hairy roots contained a lower proline content (Figure 3D). Additionally, the salt-treated OE hairy roots accumulated lower MDA levels than the control ones. However, a larger MDA level accumulated in the RNAi hairy roots than in the VC hairy roots under salt conditions (Figure 3E). Considering that the $H_2O_2$ treatment led to the increased expression of *GmCIPK2*, we analyzed the ROS contents in the hairy roots of RNAi, VC, and OE. Under normal conditions, ROS contents in the different genotype hairy roots are comparable (Figure 3F,G). Salt stress promoted ROS synthesis. $H_2O_2$ is a well-recognized moderately reactive ROS that can induce oxidative stress [1,29]. The DAB staining and quantitative assays showed that the $H_2O_2$ content was lower in the OE hairy roots than in the VC hairy roots after the salt treatment. By contrast, the salt-treated RNAi hairy roots contained higher $H_2O_2$ levels (Figure 3F,G). Antioxidant enzymes function essentially in scavenging ROS [1,3,29]. Upon salt stress, compared to the control roots, the OE hairy roots exhibited greater POD activity and GST activity, whereas the activity of

the POD and GST enzymes in RNAi hairy roots was lower (Figure 3H,I). Moreover, salt stress usually causes a $Na^+/K^+$ imbalance. The $Na^+/K^+$ ratios are negatively related to the salt tolerance of plants [1,29]. When subjected to salt stress, the OE hairy roots showed a lower $Na^+$ content, higher $K^+$ content, and lower $Na^+/K^+$ ratios than the control hairy roots. In contrast, the salt-treated RNAi hairy roots displayed a higher $Na^+$ content, lower $K^+$ content, and larger $Na^+/K^+$ ratios (Figure 3J–L).

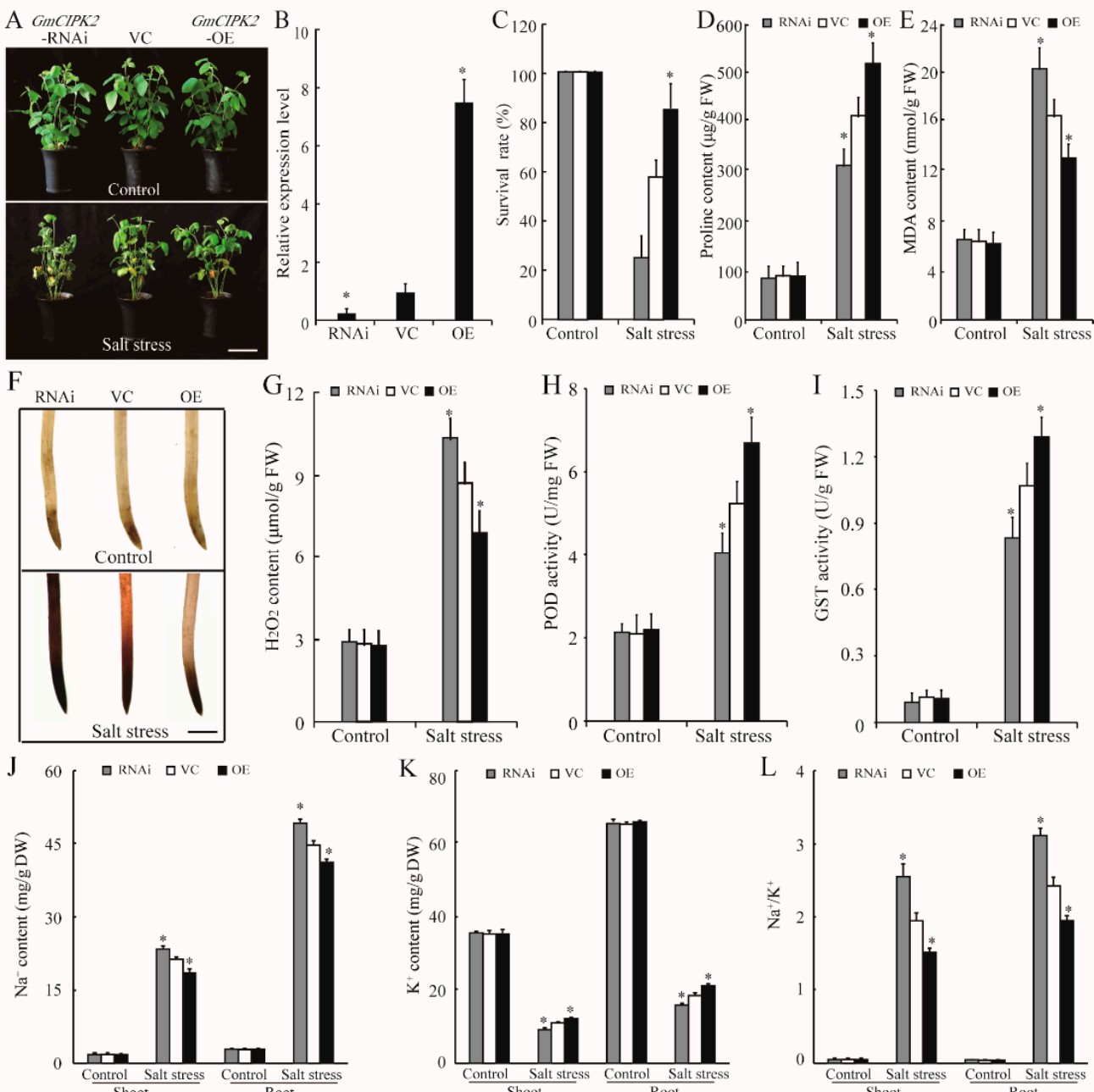

**Figure 3.** *GmCIPK2* imparts salt tolerance in hairy roots of soybean. (**A**) Analyses of the salt tolerance in RNAi, VC, and OE plants under salt treatment. Bar = 10 cm. (**B**) The transcripts of *GmCIPK2* were measured by qRT-PCR analysis. (**C**) Survival rates, (**D**) proline contents, (**E**) MDA contents, and (**F**) DAB staining of RNAi, VC, and OE hairy roots. Bar = 0.1 cm. (**G**) $H_2O_2$ content measurement, (**H**) POD activity detection, (**I**) GST activity detection, (**J**) $Na^+$ contents, (**K**) $K^+$ contents, and (**L**) $Na^+/K^+$ ratios of RNAi, VC, and OE plants under salt treatment. Each data point represents the mean (±SE) of three independent biological replicates. The * represents significant differences with the corresponding controls (* $p < 0.05$).

### 3.4. GmCIPK2 Activates the Expression of the Salt Stress- and Antioxidant-Related Genes

To clarify the molecular mechanisms of the *GmCIPK2*-mediated salt stress adaption in soybean, we examined the transcript levels of several salt- and antioxidant-related genes in the hairy roots of *GmCIPK2*-RNAi, VC and *GmCIPK2*-OE under salt treatment. No significant difference was identified among the different genotypes under normal conditions. However, when subjected to salt stress, the *GmCIPK2*-OE hairy roots showed higher expression levels of salt-responsive genes (*GmP5CS*, *GmMYB118*, *GmDHN15*, *GmLEA5*, *GmSOS1,* and *GmNHX1*) and oxidative-responsive genes (*GmPOD21*, *GmPOD47*, *GmGST18*, and *GmGST20*) than the control hairy roots. In contrast, the salt-treated *GmCIPK2*-RNAi hairy roots displayed lower expression levels of these salt- and antioxidant-related genes (Figure 4).

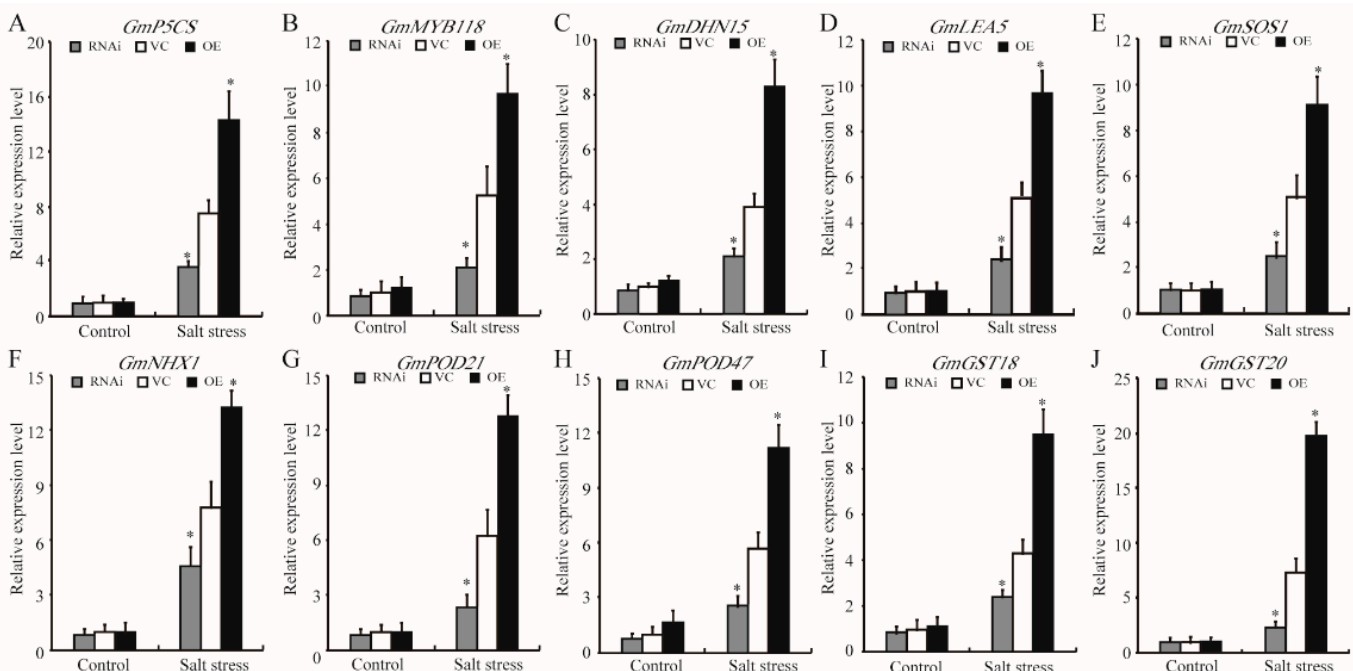

**Figure 4.** *GmCIPK2* enhances transcripts of stress-related genes regulated by *GmCIPK2*. Transcript levels of (**A**–**F**) salt-responsive gene (*GmP5CS*, *GmMYB118*, *GmDHN15*, *GmLEA5*, *GmSOS1*, and *GmNHX1*) and (**G**–**J**) antioxidant-related genes (*GmPOD21*, *GmPOD47*, *GmGST18*, and *GmGST20*) in RNAi, VC, and OE hairy roots under salt conditions. Each data point represents the mean (±SE) of three independent biological replicates. The * represents significant differences with the corresponding controls (* $p < 0.05$).

### 3.5. GmCBL4 Combines with GmCIPK2 at the Plasma Membrane

CIPKs are well-recognized for combining with specific CBLs to modulate plant adaptation to abiotic stress. The candidate CBLs that interact with GmCIPK2 were isolated using a yeast two-hybrid system. The co-expression of *GmCBL4*-AD with *GmCIPK2*-BD in the same yeast strain activated the reporter gene expression (Figure 5A). Therefore, GmCBL4 was identified as an interaction sensor of GmCIPK2.

We then performed pull-down assays to confirm the interaction between GmCIPK2 and GmCBL4. In this assay, the soluble GmCIPK2-His recombinant protein was co-purified with GmCBL4-GST but not with the control GST protein (Figure 5B), validating the direct interaction between GmCIPK2 and GmCBL4.

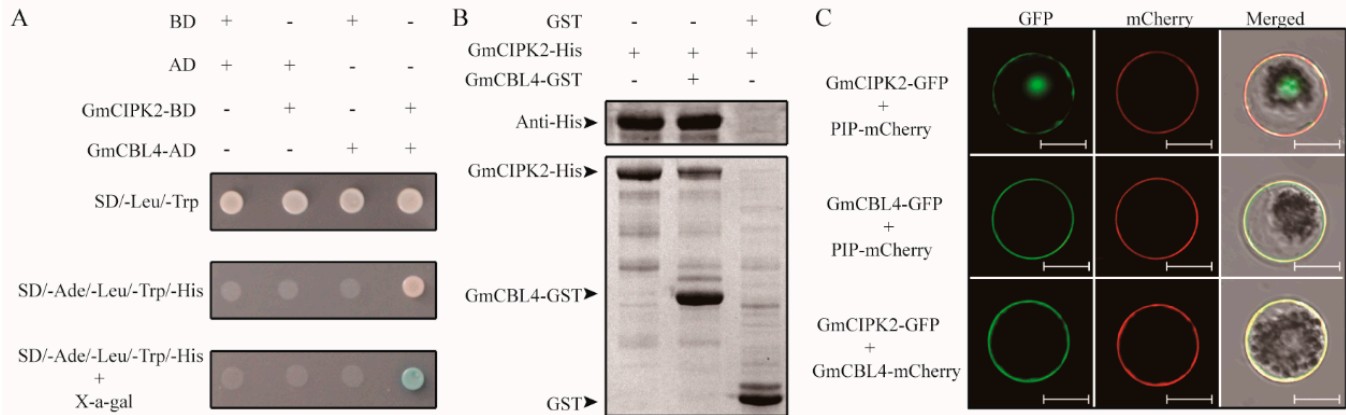

**Figure 5.** GmCBL4 is a GmCIPK2-interacting sensor. (**A**) Interaction analysis of GmCIPK2 and GmCBL4 by yeast two-hybrid assay. Transformed yeast cell (*AH109*) containing *GmCIPK2*-BD and *GmCBL4*-AD was grown in SD/-Ade/-Trp/-His/-Leu medium containing X-α-gal. (**B**) Interaction analysis of GmCIPK2 and GmCBL4 by pull-down assay. The Western blotting assay showed that GmCIPK2-His was associated with GmCBL4-GST, unlike the control GST protein. (**C**) Subcellular localization analysis of GmCIPK2 and GmCBL4 in *Arabidopsis* protoplasts. Images were observed under a laser scanning confocal microscope. Bar = 10 μm.

### 3.6. GmCBL4 Overexpression Imparts Salt Tolerance of Soybean Hairy Roots

To further explore the significance of the combination of GmCIPK2 and GmCBL4, we examined the distribution of both GmCBL4 and GmCIPK2 using the method of PEG-mediated protoplast transformation. PIP-mCherry protein was used as a membrane-localized marker [27]. GFP signals were mainly distributed in the nucleus and cytoplasm when the *PIP*-mCherry constructs were co-expressed with the *GmCIPK2*-GFP constructs in the same *Arabidopsis* protoplasts. However, as the *GmCIPK2*-GFP and *GmCBL4*-mCherry constructs were co-expressed in the same *Arabidopsis* protoplasts, GFP signals were only detected in the plasma membrane. Noteworthily, GmCBL4 encoded a membrane-localized protein (Figure 5C). Therefore, GmCBL4 may bind to GmCIPK2 at the plasma membrane to influence cellular processes.

Considering that GmCBL4 acted as a CIPK2-interacting protein, we generated hairy roots with *GmCBL4* overexpression (Figure 6B). Without salt stress, the *GmCBL4*-OE plants had a similar phenotype as the control plants (Figure 6A). The hairy roots with *GmCBL4* overexpression were more tolerant to salt stress and had significantly higher survival rates than the VC plants under salt stress (Figure 6A,C). Further physiological analyses illustrated a significantly lower MDA content and larger proline level that accumulated in the salt-treated *GmCBL4*-OE hairy roots than in the VC hairy roots in response to the salt stress (Figure 6D,E). More importantly, the results of the DAB staining and quantitative assays showed that salt-treated *GmCBL4*-OE hairy roots contained a lower content of $H_2O_2$ than VC hairy roots (Figure 6F,G). Additionally, the *GmCBL4*-OE hairy roots showed a lower $Na^+$ content, higher $K^+$ content, and lower $Na^+/K^+$ ratios than the VC hairy roots did under salt stress (Figure 6H–J).

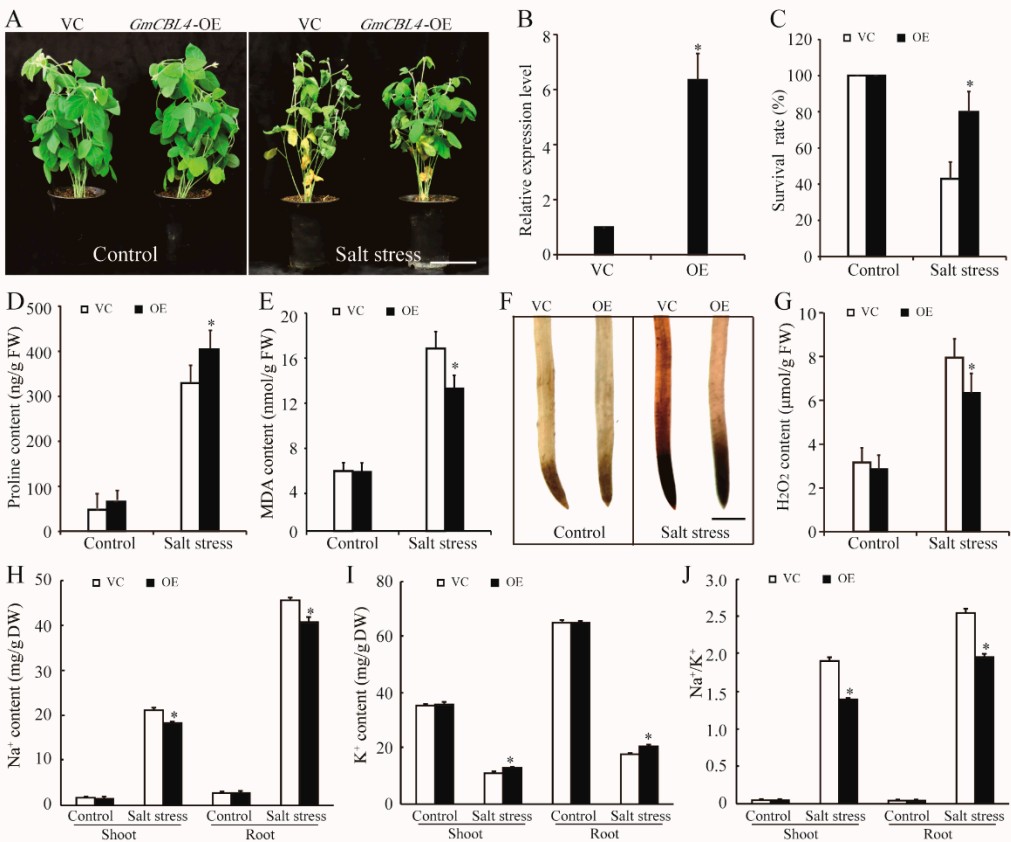

**Figure 6.** *GmCBL4* confers salt tolerance in hairy roots. (**A**) Analyses of the salt tolerance in OE and VC plants under salt treatment. (**B**) Measurement of the transcript of *GmCBL4* in hairy roots (**C**) Survival rates (**D**) proline contents, and (**E**) MDA contents of OE and VC hairy roots under salt treatment. (**F**) DAB staining and (**G**) $H_2O_2$ contents, (**H**) $Na^+$ contents, (**I**) $K^+$ contents, and (**J**) $Na^+/K^+$ ratios in the hairy roots of OE and VC plants under salt treatment. Bar = 0.1 cm. Each data point represents the mean ($\pm$SE) of three independent biological replicates. The * represents significant differences with the corresponding controls (* $p < 0.05$).

## 4. Discussion

There is ample evidence to show that CIPKs play a pivotal role in plant tolerance to salt stress [4,5]. Salt stress increases the *AcCIPK5* transcripts, and the overexpression of *AcCIPK5* confers salt tolerance in transgenic plants [34]. *ZmCIPK42* responds to salt stress, and *ZmCIPK42* overexpression leads to the improved salt resistance of maize [35]. In recent years, the whole soybean genome has been sequenced [36]; nonetheless, the functions of soybean *CIPKs* in salt stress responses remain largely unknown. In the assay, a salt-inducible gene, *GmCIPK2*, was isolated from soybean (Figure 1A). *GmCIPK2* overexpression increased the salt tolerance of *Arabidopsis* plants and soybean hairy roots. Conversely, the downregulation of *GmCIPK2* by the RNAi technology resulted in increased salt sensitivity of hairy roots (Figures 2 and 3). Taken together, *GmCIPK2* acts as a key regulator in alleviating soybean salt stress.

Salt stress has been documented to trigger the overproduction and accumulation of ROS, which causes oxidative damage to cells, such as membrane damage and enzyme activity disruption, and even cell death [1,29]. To survive, plants have evolved sophisticated antioxidant defense systems to scavenge the redundant ROS [1,17,29]. Previous studies demonstrated that *GhCIPK6a* and *BdCIPK31* increased SOD, POD, CAT, and GST enzyme activity to alleviate salt-induced oxidative stress [19,37]. In this study, the $H_2O_2$ treatment increased the transcript levels of *GmCIPK2* (Figure 1B). Further DAB staining and quantitative assays illustrated that a significantly lower content of $H_2O_2$ accumulated in the salt-treated *GmCIPK2*-OE hairy roots than in the control roots. In contrast, the oppo-

site findings were observed in the salt-treated *GmCIPK2*-RNAi hairy roots (Figure 3F,G). Furthermore, consistent with the role of *GmCIPK2* in activating the POD and GST enzymes (Figure 3H,I), *GmCIPK2* was found to promote the expression of *GmPOD21*, *GmPOD47*, *GmGST18*, and *GmGST20* under salt stress (Figure 4G–J). These results indicate that *GmCIPK2* participates in enhancing the antioxidant defense system to respond to salt stress in soybean plants.

Salt treatment increases intracellular $Na^+$ concentrations. The excessive accumulation of $Na^+$ usually triggers the inhibition of $K^+$ absorption, breaking the $Na^+/K^+$ homeostasis [1,29]. CIPKs function crucially in modulating ion transport, especially $Na^+$ and $K^+$. For example, *TaCIPK29* has been shown to enhance the expression of *SOS1*, *NHX2*, and *NHX4* to reduce the $Na^+/K^+$ ratios, improving the salt tolerance of transgenic plants [38]. *NtCIPK9* overexpression promotes the expression of *NHX1* and *NHX7* to increase the $K^+$ content and decrease the $Na^+$ content under salt stress [39]. *NHX*s encode the $Na^+/H^+$ antiporters that function in reducing the intracellular $Na^+$ content by regulating $Na^+$ extrusion or storing redundant $Na^+$ in the vacuole [40]. In this assay, upon salt stress, the overexpression of *GmCIPK2* decreased $Na^+$ concentrations, increased $K^+$ contents, and reduced $Na^+/K^+$ ratios in soybean plants (Figure 3J–L). On the contrary, the salt-treated *GmCIPK2*-RNAi lines displayed the opposite. Furthermore, *GmCIPK2* enhanced the transcript levels of *GmNHX1* and *GmSOS1* in response to salt stress (Figure 4E,F). Collectively, our findings demonstrate that *GmCIPK2* is involved in enhancing $Na^+/K^+$ homeostasis to improve the salt tolerance of soybean.

CIPKs have been reported to modulate salt-responsive gene expression to contribute to salt tolerance in plants. A previous study reported that *ZmCIPK21* increases the transcript levels of *RD29A*, *COR15*, and *DREB* to increase salt tolerance in transgenic *Arabidopsis* [14]. *NtCIPK11* was shown to regulate the expression of the proline biosynthesis-related genes to increase the proline content in tobacco under salt stress [41]. In this assay, *GmCIPK2* enhanced the transcript levels of the proline biosynthesis gene *GmP5CS* (Figure 4A), which was consistent with its positive role in increasing the proline content under salt stress (Figure 3D). Moreover, *GmCIPK2* was found to increase the transcript levels of *GmMYB118*, *GmLEA5*, and *GmDHN5* in response to salt stress (Figure 4B–D). MYB transcription factors usually function in mediating stress signal transduction to regulate plant adaption to stress conditions [29,42]. *DHN* and *LEA* encode dehydrins that play critical roles in protecting cell membrane stability, regulating ion balance, and controlling ROS homeostasis [1,3]. These findings indicate that *GmCIPK2* is associated with increasing the transcript levels of the salt-related gene, contributing to the tolerance of soybean to salt stress.

CIPKs usually combine with specific CBL proteins to regulate plant adaptation to adverse conditions [4,9]. For instance, CBL1/9 has been reported to combine with AtCIPK1 to mediate the *Arabidopsis'* adaption to osmotic and salt stresses [43]. According to a subsequent study, the AtCBL1/9-AtCIPK23 complex activates the inward $K^+$ channel AKT1 to promote $K^+$ absorption [44]. Moreover, CaCBL2 interacts with CaCIPK3 at the plasma membrane to improve the drought tolerance of transgenic tomatoes [45]. In this study, the results of the interaction assays verified that GmCBL4 functions as a GmCIPK2-interacting sensor (Figure 5). Moreover, *GmCBL4* overexpression conferred salt tolerance in transgenic hairy roots. Furthermore, compared to the control, the salt-treated *GmCBL4*-OE hairy roots had a higher proline content, a lower content of MDA and $H_2O_2$, smaller $Na^+$ content and $Na^+/K^+$ ratios, and a higher $K^+$ concentration (Figure 6), which is consistent with the function of *GmCIPK2* in improving the salt tolerance of soybean (Figure 3). Collectively, these results indicated that the GmCBL4-GmCIPK2 complex contributes to enhancing soybean salt tolerance.

## 5. Conclusions

*GmCIPK2* functions crucially in enhancing soybean tolerance to salt stress. Furthermore, *GmCIPK2* alters the antioxidant defense system, $Na^+/K^+$ homeostasis, and salt-related gene expression to respond to salt stress. Moreover, GmCBL4 functions as

a GmCIPK2-interacting sensor that improves the salt tolerance of soybean hairy roots. Overall, this study contributed to elucidating the CBL-CIPK mediated salt-responsive mechanism in soybean.

**Supplementary Materials:** The following supporting information can be downloaded at: https://www.mdpi.com/article/10.3390/agronomy12071595/s1, Figure S1. Sequence analysis of GmCIPK2. (A) Multiple sequences alignment of GmCIPK2, OsCIPK2, and AtCIPK2. Dark blue shading indicates identical residues. Dark lines demarcate the N-terminal Ser/Thr kinase domain and C-terminal regulatory domains. The NAF/FISL domain is marked with a red rectangle. Table S1. Primers used in RT-PCR assays.

**Author Contributions:** X.-Y.C. coordinated the project and conceived and designed experiments; H.L. performed experiments and wrote the manuscript; W.-L.Z., X.-H.W. and P.X. conducted the bioinformatic analysis; Z.-N.L., Q.L. and X.C. analyzed the experimental data. All authors have read and agreed to the published version of the manuscript.

**Funding:** This research was financially supported by the National Natural Science Foundation of China (No. 32001459, 32001575, 32070344) and the Natural Science Foundation of Shandong Province (No. ZR2020QC123, ZR2019PC055).

**Institutional Review Board Statement:** Not Applicable.

**Informed Consent Statement:** Not Applicable.

**Data Availability Statement:** Not Applicable.

**Conflicts of Interest:** The authors declare no conflict of interest.

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
