# Peer review of "CBL-Interacting Protein Kinase 2 Improves Salt Tolerance in Soybean (Glycine max L.)"

_agronomy, doi:10.3390/agronomy12071595_

Round 1

Reviewer 1 Report

In this work, Li and co-authors report the beneficial role of overexpressing GmCIPK2 gene in salinity stress tolerance in soybean and attribute this effect to alleviation of salt-induced ROS accumulation by CIPK2-CBL4. While the paper is potentially interesting, it suffers from some methodological flaws, and the conclusions made are not justified by the reported data. My major concerns are as follows:

11. The authors applied salinity stress for only 10 days. During this period the osmotic component of salt stress dominates, while NaCl toxicity plays a relatively minor role. The authors either need to extend the duration of salt stress or add osmotic controls to their work.

22. The conclusion about GmCIPK2 participating in scavenging salt-induced ROS accumulation is nothing by speculation. Indeed, the amount of MDA Most likely, the effect in OE lines was lower but most likely this was a result of indirect effect. There is absolutely no evidence to suggest that CIPK2 controls ROS scavenging.

33. CIPK-CBL complexes operate upstream of various Na+ and K+ transporters mediating plant ion homeostasis and, hence, salinity tolerance. To my astonishment, I have found no Na or K data in this work. This is simply not acceptable! Playing the devil’s advocate I would say that CIPK2 may reduce root Na+ uptake and/or improve K+ retention, thus enhancing plant performance under saline conditions. As these plants are less “stressed”, they will also have lower ROS accumulation and MDA content – nothing to do with ROS scavenging! Thus, tissue ion content is a MUST in this work.

44. In Fig 1, the authors report the peak in CIPK2 level after 3 h of treatment, and by 24 h its levels return back to pre-stress (controls). For me, this implies a possible signaling role of CIPK in plant ion homeostasis rather than its control of antioxidant activity – 3 h is far too early for the latter!

Author Response

Thanks for your commments! We have responsed the questions point by point, as is shown in the attachment.

Reviewer 2 Report

This manuscript focuses on the functions of GmCIPK2 and GmCBL4 in salt tolerance. I found the manuscript written in a clear manner, and different approaches support the conclusions. However, I think that the following points should be improved.

Introduction: the function of GmCIPK2 in drought tolerance was previously published by the authors. (Xu M, Li H, Liu ZN, Wang XH, Xu P, Dai SJ, Cao X, Cui XY. The soybean CBL-interacting protein kinase, GmCIPK2, positively regulates drought tolerance and ABA signaling. Plant Physiol Biochem. 2021 Oct;167:980-989. doi: 10.1016/j.plaphy.2021.09.026. Epub 2021 Sep 24. PMID: 34583133.) This reference should obviously be cited in the introduction, and the known functions of GmCIPK2 should be introduced.

Lines 77 – 83: OE and RNAi lines for GmCIPK2 were previously generated by the authors and the reference should be cited. Xu M, Li H, Liu ZN, Wang XH, Xu P, Dai SJ, Cao X, Cui XY. The soybean CBL-interacting protein kinase, GmCIPK2, positively regulates drought tolerance and ABA signaling. Plant Physiol Biochem. 2021 Oct;167:980-989. doi: 10.1016/j.plaphy.2021.09.026. Epub 2021 Sep 24. PMID: 34583133.

Besides, the vector reference should be double checked for the OE line.

Line 97: The frequency of salt treatment should be indicated. Moreover, the volume of the pots, and the volume of NaCl solution added should be indicated.

Line 102: The references of the kits should be indicated.

Line 109: “coli” should be in italic

Figure 3 A: a scale bar should be added.

Did the authors generate RNAi lines to confirm the role of GmCBL4 in the salt stress tolerance ?

Author Response

Thanks for your comments! We have responded the questions point by point, as is shown in the attachment.

Round 2

Reviewer 1 Report

The authors has ignored my request for including osmotic controls (e.g. using non-ionic solutions isotonic to NaCl used). Without such data the paper is incomplete, and any discussion remains speculative.

Author Response

Thanks for your comments! 

Reviewer’ 1:

The authors has ignored my request for including osmotic controls (e.g. using non-ionic solutions isotonic to NaCl used). Without such data the paper is incomplete, and any discussion remains speculative..

Response: Thanks! Your suggestions have provided many inspirations for us to study the mechanism of soybean respond to salt stress. High salinity usually triggers osmotic stress and ionic stress. The responsive mechanisms of osmotic stress and ionic stress are partially overlapped and very hard to separate. Several CIPKs, such as BdCIPK31 and AcCIPK5 were reported to participated in regulate osmotic stress response and Na+/K+ homeostasis. In this assay, GmCIPK2 expression was increased by NaCl and hydrogen peroxide (H2O2). GmCIPK2-overexpression Arabidopsis and soybean hairy roots displayed improved salt tolerance, whereas the RNA interference hairy roots exhibited enhanced salt sensitivity. Further analysis demonstrated GmCIPK2 was associated with increasing proline content, antioxidant enzymes activity, and stress-related genes expression and reducing MDA content and Na+/K+ ratios. Therefore, our conclusions “GmCIPK2 functions crucially in enhancing soybean tolerance to salt stress. Furthermore, GmCIPK2 alters antioxidant defense system, Na+/K+ homeostasis, and salt-related gene expression to respond to salt stress. Besides, GmCBL4 functions as a GmCIPK2-interacting sensor that improves the salt tolerance of soybean hairy roots. Overall, this study contributed to elucidating the CBL-CIPK mediated salt-responsive mechanism in soybean” are appropriate. We sincerely hope that you will support our conclusions

Reference:

  1. Aslam, M.; Greaves, J.G.; Jakada, B.H.; Fakher, B.; Wang, X.; Qin, Y. AcCIPK5, a pineapple CBL-interacting protein kinase, confers salt, osmotic and cold stress tolerance in transgenic Arabidopsis. Plant Sci. 2022, 320, 111284.
  2. Luo, Q.; Wei, Q.; Wang, R.; Zhang, Y.; Zhang, F.; He, Y.; Zhou, S.; Feng, J.; Yang, G.; He, G. BdCIPK31, a calcineurin B-like protein-interacting protein kinase, regulates plant response to drought and salt stress. Front. Plant Sci. 2017, 8, 1184.